# Clock Gene Expression Modulation by Low- and High-Intensity Exercise Regimens in Aging Mice

**DOI:** 10.3390/ijms26178739

**Published:** 2025-09-08

**Authors:** Matheus Callak Teixeira Vitorino, Hugo de Luca Corrêa, Verusca Najara de Carvalho Cunha, Mariana Saliba de Souza, Herbert Gustavo Simões, Thiago dos Santos Rosa, Elaine Vieira, Rosângela Vieira de Andrade

**Affiliations:** 1Postgraduate Program in Genomic Sciences and Biotechnology, Universidade Católica de Brasília, Brasília 71966-700, DF, Brazil; matheuscallak@outlook.com (M.C.T.V.); vieiraec@yahoo.com (E.V.); 2Postgraduate Program in Physical Education, Universidade Católica de Brasília, Brasília 71966-700, DF, Brazil; hugo.efucb@gmail.com (H.d.L.C.); najavrusk@gmail.com (V.N.d.C.C.); hgsimoes@gmail.com (H.G.S.); thiagoacsdkp@yahoo.com.br (T.d.S.R.); 3Faculty of Science and Health, Centro Universitário de Brasília, Brasília 70790-075, DF, Brazil; marianasaliba662@gmail.com; 4Faculty of Education, Universidade Estadual de Minas Gerais, Passos 37902-407, MG, Brazil

**Keywords:** circadian cycle, chronic exercise, swimming, heart, skeletal muscle, clock genes

## Abstract

The circadian rhythm controls the sleep/wake cycle and a wide variety of metabolic and physiological functions. Clock genes regulate it in response to both external and endogenous stimuli, and their expression may change because of aging, leading to an increased risk of health problems. Despite the well-described benefits of physical exercise as a circadian synchronizer, there is a lack of literature regarding the role of chronic exercise intensity in clock gene expression during aging. This article aims to analyze the differential expression of genes that regulate the biological clock under the effects of variable-intensity aerobic swimming training in aging mice, determining whether these exercise regimens interfere with the genomic regulation of the circadian rhythm. For this purpose, the mice were exposed to low- and high-intensity exercise and had their heart and gastrocnemius tissues molecularly analyzed by cDNA synthesis and qPCR to determine the expression levels of the selected genes: *Clock*, *Arntl*, *Per1*, *Per2*, *Cry1*, *Cry2*, and *Nr1d1*. The results showed that low-intensity exercise, performed at workloads below the anaerobic threshold, significantly changed their expression in the gastrocnemius muscle (*p* < 0.05), while high-intensity exercise had no statistically significant effects (*p* > 0.05), with the heart being immune to exercise influence except when it comes to the *Per1* gene, for which expression was increased (*p* = 0.031) by low-intensity exercise. Additionally, both body weight and lactate thresholds had no change during the experiment (*p* > 0.05), while the maximum supported workload was maintained for high-intensity exercise (*p* > 0.05) and increased for low-intensity exercise (*p* < 0.01), with the control group experiencing a decay instead (*p* < 0.05). Thus, the present study highlights the importance of chronic exercise in modulating clock genes and opens exciting possibilities for circadian medicine, such as improvements in exercise capacity, heart condition, and lipid metabolism for subjects of low-intensity regimens.

## 1. Introduction

The circadian rhythm controls the sleep/wake cycle and metabolism and repeats every 24 h, involves chemical, physiological, and behavioral events essential for survival [1], and can be observed in plants, animals, fungi, and bacteria [2]. Both its value for survival and the presence of the mechanism in a reasonably conserved form, with different genes but common parts such as peroxiredoxin protein oxidation–reduction, in the most diverse types of living beings are due to the ability to anticipate and prepare for regular changes in the environment that it offers [2]. Some examples of this are plants opening their leaves during the day to absorb light and closing them during the night to conserve energy [3], as well as animals aligning their feeding, mating, and activity patterns with the most favorable times to avoid predators and optimize resource use [4].

The cycle is regulated by external stimuli, such as light and temperature, and endogenous stimuli, under the mechanism known as the biological clock [5], which consists of the central oscillator (suprachiasmatic nucleus—SCN—of the hypothalamus) and peripheral oscillators (present in the adrenals, esophagus, lungs, liver, spleen, thymus, and skin). The central oscillator receives light stimuli detected by the retinas and synchronizes the peripheral ones, which are semi-independent until they receive a superior signal, having their own cycles of approximately 24 h, controlling the metabolism and other activities of the organism [6,7].

The genomic control of the circadian rhythm in mammals occurs through the intracellular transcription and translation feedback cycle, divided between the primary and secondary systems [5]. The primary system is composed of two opposing regulatory arms: one positive, or stimulatory, composed mainly of the circadian locomotor output cycle kaput (*Clock*) and aryl hydrocarbon nuclear receptor translocator-like protein (*Arntl*), also known as Brain and muscle arnt-like1 (*Bmal1*), genes, and the other negative, or repressor, composed mainly of the Period (*Per1*, *Per2*, *Per3*) and Cryptochrome (*Cry1*, *Cry2*) genes. The first arm involves the heterodimeric transcription factor CLOCK + ARNTL, which activates the expression of the genes of the second arm through binding with their promoter regions. During the active period, the levels of the negative arm rise until they enter the nucleus and inhibit the action of the positive arm through binding with the CLOCK + ARNTL dimer itself. When the components of the negative arm are degraded by external stimulation, the positive arm regains its transcriptional activity, restarting the cycle. The positive arm predominates during the day, causing wakefulness, while the negative arm predominates during the night, causing sleep [5], with phase changes in the peripheral oscillators of diurnal and nocturnal organisms generating their differences in behavior [8].

The secondary system, in turn, is composed of three groups: the NUCLEAR RECEPTOR SUBFAMILY 1 GROUP D MEMBER 1 (NR1D1), also known as REV-ERB, and RETINOIC ACID-RELATED ORPHAN RECEPTOR (RORα, RORβ and RORγ) receptors, which regulate *Arntl*; the D-ELEMENT BINDING PROTEIN (DBP) and E4 BINDING PROTEIN 4 (E4BP4) proteins, which bind to the *Per* promoter sequence and alter its expression; and Deleted in esophageal cancer (Dec1, Dec2), stimulated by CLOCK + ARNTL, whose products are repressors of *Per* and self-targeted receptors [5].

Post-translational modifications can also interfere with the cycle through the phosphorylation or dephosphorylation of clock proteins. This can be performed by CASEIN KINASE 1 (CK1ε and CK1δ), which facilitates the degradation and entry into the nucleus of PER2, CASEIN KINASE 2 (CK2), which mobilizes PER to the nucleus, and PROTEIN PHOSPHATASE 1 (PP1), which dephosphorylates PER2, stabilizing it for re-entry into the nucleus [5,6].

The main clock genes also have other functions in the organism:

*Clock*, for instance, in addition to its role in the genomic regulation of the biological clock as a *Per* and *Cry* transcription factor [5], regulates physical capacity for exercise [9,10]. Mutations in the gene can generate longer or arrhythmic circadian cycles [11]. In the heart, the cycle is predominantly controlled by the central oscillator, with variations in *Clock* expression not modifying the circadian rhythm but increasing the risk of myocardial infarction [12] and reducing physical aptitude [9], which can, however, be partially restored through the practice of exercise [10].

*Arntl*, similarly to *Clock*, aside from being a *Per* and *Cry* transcription factor [5], controls bone mass through mesenchymal stem cells [13] and prevents myocardial infarction [12] by regulating blood pressure [14], plasma lipid levels, and the gene expression profile of perivascular adipose tissue and its macroscopic function [15]. It also controls the transcription of genes involved in metabolic inflammation and adipose tissue remodeling [16], which are involved in the regulation of half of all animal genes [14].

*Per*, the *Arntl* and *Clock* inhibitor, negatively mediates toxin transcription [17], serves as a diabetic aggravation marker [18], and helps with hippocampal memory processes [19,20]. The gene is regulated by the glucocorticoid REDD1 signaling pathway in the skeletal muscle after acute aerobic exercise [21].

*Cry*, meanwhile, sharing the function of *Per* in the circadian regulation, has its subtype *Cry1* that stimulates tumorigenesis [22], while *Cry2* is involved in DNA repair [23]. It also influences stress and depression responses [24] and regulates testicular function, cell communication, chromatin reorganization, spermatogenesis, and immune response [25].

*Nr1d1*, finally, which is highly expressed in the adipose tissue, skeletal muscle, heart, liver, and brain [26], in addition to its role in the genomic regulation of the biological clock as a repressor of *Arntl* [5,27], also regulates cell proliferation and differentiation [26], tumor suppression [28], lipid and carbohydrate metabolism [27], and inflammation [29]. Its reduced or null expression can shorten the circadian cycle and create aberrant responses to the external regulation of the light oscillator [27]. It is indispensable for adipogenesis, with increased expression enhancing it [27] and helping in adipocyte differentiation [26,30]. Its metabolic effects range from lipid metabolism, by repressing the production of apolipoprotein CIII, which raises the blood levels of triglycerides and VLDL in the case of failure, passing through carbohydrates, by repressing the gluconeogenic enzyme glucose 6-phosphatase, and reaching the metabolism of bile acids [27]. It also links the lung clock to innate immunity, regulating the inflammatory response to determine the threshold between healthy and inflamed tissue states, with its lower expression creating a baseline state of permanent inflammation [29].

Disruption in the circadian rhythm can cause several health problems, such as hyperphagia, obesity, alterations in the glucose metabolism by mutations in *Clock* [31], an imbalance in the use of glucose and lipids, predisposing mice to diabetes, by the deletion of *Nr1d1* [32], and metabolic syndrome and cardiovascular problems in humans by polymorphisms in the CLOCK gene [33]. The effects of the Circadian Syndrome can be attenuated by adjusting exercise time, light exposure, food consumption, medication use, and sleep with circadian medicine [34].

Aging and the circadian cycle have a mutually disruptive relationship [35]. This means aging impairs clock gene expression, causing circadian imbalances. These imbalances, whether due to old age or factors like late-night shift work [36], in turn, accelerate the aging process, worsening the cycle over time. This disruption can lead to diseases such as breast and prostate cancer [36], as well as Alzheimer’s and Parkinson’s diseases [35]. Even at the cellular level, the circadian clock, which plays a crucial role in regulating the cell cycle [36], is adversely affected by aging. Senescent cells lose the expression of the clock genes, impairing the tissue’s ability to maintain circadian regulation. This impairment persists even when older cells are implanted into younger tissues, indicating that it is independent of external stimuli attempting to repair the issue [37].

Physical exercise is a synchronizer of the skeletal muscle peripheral oscillator in humans [38], with factors such as intensity, duration, time, and the presence or absence of light being tested to determine its effects on individuals of different ages by an initial observation in rodents, then in humans, with experiments that nullify the influence of other photic and non-photic synchronizers, creating constant situations for routine protocols [39]. Night exercise [40] and its different intensities and durations [41], for example, alter the rhythms of thyrotropin and melatonin the following day, with high-intensity exercises for periods of 60 min being ideal for application in humans in a normal situation [41], whose greater power, strength, and endurance are usually observed between the afternoon and night [38], in addition to the ideal core, body, and skeletal muscle temperatures [42]. Morning exercise, in turn, can advance the clock phase [43].

The literature is sparse in articles that correlates the expression of the selected genes with chronic exercise, with most experiments involving its acute effects, which presents a need for more publications and perspectives.

This article, as the first study on the effects of a prolonged regimen of variable-intensity swimming exercise on aging mice, aims to analyze the differential expression of genes that regulate the biological clock under the effects of low- and high-intensity aerobic swimming training in aging mice, determining whether these exercise regimens interfere with the genomic regulation of the circadian rhythm.

## 2. Results

To characterize the functional groups of clock genes, which are transcriptionally regulated in an intensity-dependent manner by exercise, a gene expression analysis by qRT-PCR was conducted in the heart and white gastrocnemius muscle. Statistically significant changes in expression are represented by a *p*-value of *p* < 0.05.

No significant changes in *Arntl*, *Clock*, *Cry1*, *Cry2*, *Per2*, and *Nr1d1* expressions were detected in the heart (Figure 1A–D,F,G; *p* > 0.05). Meanwhile, low-intensity exercise increased the mRNA expression of the *Per1* gene in the cardiac muscle compared to the control group (Figure 1E; *p* = 0.031), but high-intensity exercise did not have significant effects (Figure 1E; *p* = 0.37).

The expression pattern in the gastrocnemius muscle was different from the heart, with *Arntl* (Figure 2A; *p* < 0.0001) and *Cry1* (Figure 2C; Ctrl × Low: *p* = 0.002/Ctrl × High: *p* = 0.032) reducing their expression by exercise independent of the intensity. *Cry2* mRNA expression was also decreased by exercise compared to the control group but only under low-intensity conditions (Figure 2D; *p* = 0.005). *Nr1d1*, as expected, being a negative regulator of *Arntl* [27], had significantly higher mRNA expression in both low- and high-intensity exercise groups compared to the control group (Figure 2G; Ctrl × Low: *p* = 0.002/Ctrl × High: *p* = 0.022).

These results indicate that low-intensity exercise has a greater impact on gastrocnemius gene expression, with minimal effects observed in the heart.

A heatmap analysis further reinforces the sensitivity of the genes to low-intensity exercise (Figure 3).

Meanwhile, Figure 4 illustrates the associations between genes and how they are vectored to the exercise protocols (Figure 4A,B) and the contribution of each gene for the total variance model (Figure 4C,D).

Additionally, body weight did not present statistically relevant differences between the groups and was maintained during the experiment (control: 31.25 ± 2.06 g; low: 30.75 ± 1.26 g; high: 31.75 ± 3.10 g; *p* > 0.05). Lactate thresholds also did not present any considerable changes between groups and during the measurements (*p* > 0.05). The feeding patterns of the mice showed discrete increases in the average consumed rations towards the end of the period for all groups (*p* < 0.05), especially for the sedentary mice.

Fasting blood glucose was increased in the Ctrl group (121.5 ± 3.6 mg/dL to 133.6 ± 3.8 mg/dL, *p* < 0.05), showed a slight reduction in the high group (128.9 ± 6.3 mg/dL to 121.7 ± 6.7 mg/dL, *p* > 0.05), and was improved significantly in the low group (140.9 ± 6.2 mg/dL to 112.7 ± 10.9 mg/dL, *p* < 0.05). The decrease in the high group is not statistically significant. The GLUT4 content of gastrocnemius cell membranes was increased, compared to the Ctrl group (~0.6 ± 0.05), of approximately 66.7% for the low group (~1 ± 0.2, *p* = 0.008) and 133.3% for the high group (~1.4 ± 0.1, *p* = 0.001).

Maximum supported workloads were the same at the start for all groups (Ctrl: 1.24 ± 0.2 g; high: 1.35 ± 0.15 g; low: 1.08 ± 0.17 g; *p* > 0.05), and then they decayed in the control group (0.91 ± 0.2 g; *p* < 0.05), while they were either maintained (low: 1.13 ± 0.1 g; *p* > 0.05) or raised (high: 1.54 g ± 0.1 g; *p* < 0.01) in the low- and high-intensity groups.

The heart function, estimated by echocardiogram readings, seemed to be increased in both exercise groups, with a loss in sedentary mice.

The ejection fraction (EF%) were similar at the start for all groups (Ctrl: 61 ± 0.5%; high: 60 ± 0.5%; low: 60 ± 0.3%; *p* > 0.05), and then it was reduced in the control group (58 ± 0.6%, *p* < 0.05) and increased in both the interference groups (low: 65 ± 1%; high: 68 ± 0.8%, *p* < 0.05). The volume of each ejection was also increased in both the low- and high-intensity groups (low: 25 ± 1 µL to 30 ± 2 µL; high: 27 ± 2 µL to 34 ± 1 µL, *p* < 0.05), while the sedentary mice remained stable, albeit with a discrete reduction (24 ± 0.6 µL to 23 ± 0.9 µL, *p* > 0.05).

Contraction potential was measured by fractional shortening (FS%), which was stable at the start in all groups (Ctrl: 33 ± 2%; low: 32 ± 3%; high: 31 ± 1%, *p* > 0.05). After the regimen, the control group showed a slight loss of contractibility (30 ± 1%, *p* < 0.05), while the exercise groups showed a gain instead (low: 35 ± 2%; high: 40 ± 3%, *p* < 0.05). The resting heart rate, meanwhile, varied upwards, but with no statistical significance, for the non-exercised mice (525 ± 25 bpm to 550 ± 30 bpm, *p* > 0.05), while it varied downwards in the low- (530 ± 20 bpm to 502 ± 22 bpm, *p* > 0.05) and the high-intensity groups (533 ± 30 bpm to 480 ± 22 bpm, *p* < 0.05), with only the high-intensity group showing statistical significance.

Finally, for the left ventricular dimensions during diastole, both wall thickness and diameter remained statistically stable among all groups before and after exercise (Ctrl: 0.95 ± 0.1 mm thick and 3.1 ± 0.4 mm wide to 0.95 ± 0.12 mm thick and 2.9 ± 0.3 mm wide; low: 0.9 ± 0.12 mm thick and 3.0 ± 0.5 mm wide to 0.92 ± 0.13 mm thick and 3.1 ± 0.2 mm wide; and high: 0.92 ± 0.09 mm thick and 3.2 ± 0.3 mm wide to 0.97 ± 0.1 mm thick and 3.5 ± 0.2 mm wide, *p* > 0.05).

## 3. Discussion

This experiment enforced both high- and low-intensity chronic exercises to assess their interfering power on the expression of the main clock genes in the gastrocnemius muscle and heart of mice to understand the varied effects that exercise can have on different tissues under different exercise regimens. *Nr1d1*, *Arntl*, *Clock*, *Per*, and *Cry* were selected as genes of interest, accurately representing both the primary and secondary genomic regulation systems of the circadian rhythm, in addition to using the antagonism between two components as a guarantee of data quality.

In the heart, only the *Per1* gene had its expression changed, an expected response to stress, unrelated to the exercise itself, with an increase being observed exclusively on the low-intensity regimen, being more stressful due to its longer duration.

The unique behavior of *Per1* in the heart warrants a further literature exploration regarding its knockout effects. Mice lacking the *Per1* gene exhibit altered circadian rhythm and impaired heart function, with an increased likelihood of hypertension and kidney injury when on a sodium-rich diet [44]. This can lead to cardiovascular issues and diminished fitness levels, partly due to the decreased production of endothelin-1 (ET-1), a critical vascular regulator [44]. Additionally, *Per1* regulates the body’s response to physical activity. Its knockout reduces metabolic response and exercise capacity, negatively impacting health and fitness [44]. Indirect inhibition, such as knocking out the beta-2 adrenergic receptor (β2-AR), which induces *Per1* expression, can also result in the loss of *Per1* function, leading to similar effects [45], implying that adrenergic signaling is related to *Per1* and may be affected by its knockout as well.

In the gastrocnemius muscle, a stress-resistant tissue, however, several genes had their expression altered. *Arntl* and *Cry1* had their expression reduced for both intensities, as did *Cry2*, although only in the low-intensity regimen, while *Nr1d1*, as expected, being a negative regulator of *Arntl*, was upregulated in both intensities.

*Clock* remained stable in both tissues under both regimens, which shows an independence of this important gene in the regulation of the circadian cycle and other body functions.

Additional physiological metrics were analyzed to complement the gene expression findings:

Body weight seemed to not be impacted whatsoever, which makes sense considering the slight increase in food intake that all groups showed for compensating for the exercise.

Cardiac function, meanwhile, showed slight degeneration in the sedentary group, with exercise clearly having a positive impact on all measurements except the heart dimensions, although the high group showed slight impact on the hear dimensions. The high group had a higher effect on the cardiac function, which translates, as expected, into more exercise capacity. The low group, meanwhile, also had more discrete improvement in heart function accompanied by the maintenance of their supported workloads. Ctrl followed the heart degeneration by a constant loss of physical fitness.

The regulation of carbohydrate metabolism is also a known function of the biological clock [46], with the results presented here being quite agreeable with the clock gene expression findings: GLUT4, an essential gatekeeper for glucose intake in cell membranes of insulin-sensitive tissues [47], was increased in both exercise groups in the gastrocnemius muscle along with an improvement in basal glycemia, signaling that exercise and its effects on clock gene expression had a direct effect over the maintenance of this facet of homeostasis. The direct correlation with clock gene regulation is reinforced by the fact that the biggest decrease in glucose levels was observed in the low group, the one that also had the biggest effect on the regulation of clock gene transcription.

After these complementary observations, it is important to discuss gene function analysis. It can involve the following steps: examining transcriptome data to assess mRNA availability after intervention, measuring proteome changes to determine actual protein production, and evaluating phosphorylation for functional protein modulation. Changes in mRNA levels typically lead to phosphorylation shifts that occur rapidly and align closely with transcriptomic changes as well as delayed and variable changes in protein abundance [48]. This delay is caused by the time it takes for the mRNA to be translated, while the variation comes from three different factors: miRNA interference, ribosomal availability, and protein stability [49]. miRNAs, such as miR-34a [50] and miR-204-3-3p [51], impede clock gene expression, especially when exercise is chronic, thus reducing their function and enhancing their translational capabilities [52]. Ribosome activity is another marker of translational capability, and exercise can increase the production of their components [53], biogenesis, and activity [54], enhancing their function. Finally, protein stability protects proteins from degradation and increases their half-life [55], with exercise increasing it [56] either by the downregulation of degradation mechanisms [57], a higher protein turnover [58,59], or the enhanced production of chaperones such as the Heat Shock Proteins (HSPs) [60,61]. Thus, exercise is a natural counter to the factors that negatively affect the connection between the transcriptome and the proteome in the skeletal muscle, making mRNA level shifts have greater impact across omics layers. The additional advantage that housekeeping genes are more stable [62] than housekeeping proteins during exercise and between different individuals and tissues [63,64] makes transcriptome studies easier to standardize under the current conditions, convincing the team to focus on the scarce available samples for this approach. This further reinforces the fact that circadian proteins usually have a lower overall abundance and are harder to detect [65].

This project contrasts with most methods described in the current literature, comprising single-bout, or acute, exercise [21,66,67,68,69], single or variable intensity [66,67,68,69,70,71], single tissue [67,68,69,70], few genes [66,67], voluntary work [71], and different species [67,68,69,70]. The most similar work to this study was led by Erickson et al.: elderly humans were exposed to a 12-week exercise regimen of variable intensity for improvements in body composition, insulin sensitivity, and exercise capacity, as well as an upregulation in ARNTL and PER2, with CLOCK, PER1, CRY1, and CRY2 not suffering any alterations [72]. However, fundamental differences observed in the species studied and a lack of control of exercise intensity led to differing results, except for CLOCK and PER1, which behaved similarly to their mice homologs. Due to the novel and comprehensive nature of the current experiment, it is unfortunately not possible to adequately compare results with those present in the rest of the literature, although there are noteworthy projects that deserve to be mentioned.

Mice present a considerable difference in clock gene expression between younger and older specimens [71]. In single-bout low-intensity exercise of the skeletal muscle, they show upregulated *Nr1d1* [66], *Per1*, and *Per2* [21], similar to the current findings, except for the *Per* genes, due to the stimulation being only acute. When this single bout is higher in intensity, *Arntl* is upregulated, while the *Per* genes suffer no influence whatsoever [21]. When the regimen is voluntary and chronic (albeit a bit too short), with variable intensity, meanwhile, the heart shows immunity to exercise regulation, confirming the current findings of *Per2* modulation in the gastrocnemius muscle and *Clock* suppression [71], possibly being targeted instead of *Arntl*, but leading to similar effects due to the lower generation of the CLOCK + ARNTL dimer. This difference in target within the duo could be due to the shorter period of the exercise regimen, and it was not enforced upon the subjects, along with its intensity not being controlled.

When it comes to humans, except for Erickson’s experiment [72] mentioned above, the others have a predilection for single-bout exercise and have presented interesting findings, such as leukocytes [67] respond to exercise by imitating the quadricep skeletal muscle [69], raising the expression of ARNTL and CRY1, while tendons are completely immune to either exercise or immobilization regulating its clock genes [68].

Turning the sight back to the current work, reinforced by data clustering and the understanding of the functions and relations between the genes of interest, as well as the maintenance/enhancement of physical capacity between exercised and sedentary mice, it becomes evident that lower-intensity chronic exercise has a major effect on the expression profiles of the circadian genes in the gastrocnemius muscle, while the heart resists regulation in both regimens.

These different responses come from the fact that, depending on the tissue selected for the experiment, expression profiles can vary wildly. This is due to fundamental differences between both muscle tissues: the heart, composed of type I and IIa fibers, has higher oxidation reliance, depending less on glycolysis [73], with the reverse being true, however, for the disease [74]. The gastrocnemius muscle, meanwhile, is composed of type IIb fibers and has higher glycolysis reliance [75,76,77]. Additionally, the heart has more oxidative phosphorylation capacity and mitochondrial density than the skeletal muscle [78], and their mitochondria are morphologically and biochemically different due to the discrepancy in muscle-fiber-type composition between them [79,80]. Specifically for the skeletal muscle, *Nr1d1* regulates mitochondrial biogenesis and function [66,81], with its deficiency leading to reduced mitochondrial content and oxidative function, resulting in compromised exercise capacity [82]. Long-duration exercise, however, can lead to higher quantity and quality of mitochondria locally [79,83]. Thus, with such differences in metabolism and mitochondrial content, it is expected that both tissues will react differently to the experiment.

Indeed, these unique fiber-type compositions justify the prevalence of one type of exercise over the other in the gene regulation of each muscle, since different intensities of exercise also recruit different types of muscle fibers, with high-intensity or -resistance exercise recruiting oxidative or glycolytic type II (oxidative—IIa; glycolytic—IIb), or fast, fibers and low-intensity or -endurance exercise recruiting type I (oxidative), or slow, fibers [84]. What differentiates these exercise intensities is the anaerobic threshold, with high-intensity exercise running above it and exerting a higher level of physiological stress due to the increased anaerobic contribution for ATP resynthesis during muscle contractions and low-intensity exercise achieving the opposite, which applies to both humans [85,86,87,88,89] and mice [90], with the exercise that runs above the anaerobic threshold, both acute and chronic, usually being associated with more expressive physiological adaptations [87,88]. Interestingly, even though high-intensity exercise tends to be the choice as a treatment for several chronic conditions, such as hypertension, diabetes, obesity, and inflammatory diseases [91,92], the low-intensity exercise seems to have a stronger effect on the expression of the clock genes. This is an important find that leads to the conclusion that lower intensities might be more adequate when treating a disease that involves the circadian cycle or the secondary functions of the clock genes, such as carbohydrate/lipid metabolism and inflammation, with exercise.

Stress is also an important influence over the peripheral oscillators. Its effects are time-of-day-dependent, with more detrimental results in mice during the night [93]. This is due to internal stressors warranting responses during the active phase, while external stress involves reactions during the inactive phase, conditioning each type to have more of an effect over the subject in the respective predominant time, while the opposite phase presents resistance instead [94]. Stress can also be classified by its frequency, with acute stress affecting non-genomic signaling, especially glucocorticoids and epinephrine, and chronic stress favoring genomic signaling [93] and, when enforced over longer periods of time, having less effect over the regulation of the clock due to habituation [93]. As an example, rats exposed to stress at the end of the light phase over twelve weeks show no change in circadian activity, body temperature, or feeding and drinking behavior [95]. Whatever the type, the main areas that have their clock gene expression regulated by stress are brain regions such as the amygdala, the nucleus accumbens, and the prefrontal cortex [19,94], but others such as the heart, liver, preadipocytes, kidneys, bronchial epithelial cells, pancreas, bone tissue, cornea, and fibroblasts can also be affected, especially *Per1*/2 [96] in forced-swimming exercise regimens [97], which was confirmed in the current experiment. The skeletal muscle, however, seems resistant to external stress regulation [98], but not to exercise-induced stress [99], which has an effect both in the active and inactive periods of the subject [100]. Finally, C57BL/6 mice are more resistant to chronic mild stress influencing their clock gene expression in longer regimens of weekly exercise with 3 days per week of activity [101], i.e., the same method applied here. The choices made in this experiment—the type of mice and tissue and the timing, frequency, and duration of the exercise—all contribute to reduce the effects of stress over the genomic regulation of the circadian cycle to a minimum and strengthen the results’ relation to exercise, making an in-depth analysis of other stressors unnecessary, especially after Cunha et al. related the telomere length to stress in a previous experiment with the same mice, explaining their stress profiles well enough [102].

The expression of all clock genes in aging mice is imbalanced, demonstrating the degeneration of the biological clock control and other functions in their local tissues over time [71]. With the number of distinct functions that the genes of interest have, this decline in expression over the course of life can be devastating for an organism.

This can be countered, however, with exercise. Clock gene expression, as well as cholesterol and fatty acid metabolism, are improved between sedentary and wheel running mice [103,104] through mechanisms such as stimulation by stress hormones such as aldosterone and epinephrine [21]. Even just mimicking muscular contractions pharmacologically has similar effects over the expression of *Per2*, raising it by the activation of its promoter. Performing the opposite action, as expected, inverts the effect [105].

With important factors for homeostasis such as heart function, measured by markers such as the ejection and shortening fractions, being altered in mice with disruptions in the circadian clock [106], it is useful that scheduled exercise has the potential to modulate daily rhythms and counteract aging and diseases due to the circadian system [107], serving as a potential treatment for diverse clinical conditions.

Thus, exercise, especially that of low-intensity, which consistently modulates the expression of the clock genes in aging specimens, has the positive effect of recovering, at least partially, the age-degenerated regulation of the circadian cycle and all involved mechanisms for improved homeostasis and metabolism, essential for healthy aging.

These findings, despite having been discovered in mice, are replicable in humans, since the regulation of the circadian cycle is extremely conserved between the two species [108]. This is evident in the improvement in metabolic factors, such as peripheral insulin sensitivity, suppression of insulin-mediated adipose tissue lipolysis, fasting glucose, exercise performance, fat mass, and basal liver glucose output, in diabetic men after afternoon exercise [109], as well as the regulation of local circadian rhythm after resistance exercise [69]. These factors, in face of a disturbed biological clock, would tend towards glucose intolerance, higher energy expenditure, fasting hyperglycemia [110], and insulin resistance [108], with a partial reversal through exercise [110] demonstrating correlation with the results obtained in rodents. This is especially important since circadian aging is also very damaging to humans [111].

The extrapolation of these results to humans is of utmost importance since it is known that exercise influences diverse facets of their homeostasis, for example, the maintenance of telomeres [101], the prevention of metabolic syndrome [34], and the non-genomic regulation of the circadian cycle [38]. The key role of exercise in the genomic regulation of the biological clock is now evident, since its practice, especially at more moderate intensities and in older individuals, can modify the expression of its regulatory genes, confirming the hypothesis of the experiment.

With the project being a follow-up to another group’s experiments [101], samples other than the heart and the gastrocnemius muscles, as well as the animals themselves, were not accessible. This limitation hindered further behavioral/systemic analysis and valuable exploratory avenues such as the brain (SCN), liver, and blood analysis. Further validation of the results by Western blotting and/or immunohistochemistry was not possible, but proteome analysis through publicly available data and bibliography was explored to compensate for this shortcoming. The experiment also included a male-only cohort with a restricted sample size, further imposing limitations. Despite these constraints, the group managed to find statistically significant results that contribute to elucidating the mechanisms underlying the biological clock and its regulation by exercise. Subsequent investigations in the subject will be undertaken by the team, and other researchers are encouraged to contribute to future studies aimed at addressing the constraints outlined above.

For this purpose, further enriching the knowledge about this field with more publications on the subject is paramount, given the scarcity of articles on the genes of interest. The correlation of physical exercise with the maintenance of the organism is a broad and multidisciplinary area that should offer advances in the coming years, improving the understanding of health and diseases and enabling new mechanisms to improve quality of life. Interesting areas for further investigation are the correlations of the immune system with the circadian clock, exercise and aging, life-long exercise effects on circadian health, the use of exercise as a non-pharmacological treatment for circadian disorders, and the further associations of exercise, in general, with the biological clock.

## 4. Materials and Methods

**Specimens and experimental groups:** The initial project was approved by the Universidade de Brasília’s animal research ethics committee (UnBDOC no. 29299/2009 and 81504/2010) and by the Universidade Católica de Brasília’s animal use ethics committee (protocol no. 024/14) and performed in agreement with the precepts of the law no. 11.794 of 8 October 2008, the Decree 6.899 of 15 July 2009, and complimentary regulations, as well as following the animal experimentation ethical principles adopted by the Conselho de Controle de Experimentação Animal (CONCEA) and the ARRIVE guidelines.

To carry out this project, 12 aging (12-month-old) male mice (C57BL/6) were selected, divided randomly into three groups, i.e., control (sedentary), low (low-intensity exercise), and high (high-intensity exercise), with four specimens each, and trained [112]. The animal models, produced from their parents in the Universidade Católica de Brasília’s bioassay lab, were kept in the bioterium of Universidade Catolica de Brasília’s physical education and health studies laboratory in collective cages with an approximate temperature of 22 °C, with a 12 h light/dark cycle and free feeding (Novital) and hydrating.

**Exercise regimen:** The animals were first adapted to the liquid medium through the maintenance of contact with water in an aquarium specially designed for swimming, with a controlled average temperature of 30 ± 2 °C, used in all the other steps of the experiment, during three weeks, three days per week, for 20 min, with the purpose of reducing the animals’ stress during the swimming exercise [112].

The functional evaluation of the animals was performed by means of an incremental swimming test to determine individual maximum supported load, starting with 1% of the specimen’s body weight and adding more with each 3 min of exercise until exhaustion [113], with 1 min pauses for changing the load, tied to the proximal portion of the animal’s tail with Micropore^®^ (Micropore Technologies, Redcar, UK). This test was repeated each four weeks to adjust loads and determine physical capacity.

The threshold between high- and low-intensity exercises was set at 4.5% of the specimen’s body weight, with 3% corresponding to low-intensity exercise and 6% corresponding to high-intensity exercise [112]. The loads were determined by blood lactate response tests, demonstrating that both groups have drastically different profiles of metabolic stress [112]. Repetitions of these tests were made to keep track of changes in the lactate thresholds of the animals.

Finally, specimens were subjected to the exercise regimens described above, with both groups being subjected to twelve weeks of swimming training, three times per week, always in the morning, and precisely at 8 AM. The timing and frequency of the exercise are important to familiarize the specimens and reduce the stress impact over the results [100], and they were shown to have the biggest impact on clock gene expression [48]. The low group had 40-min sessions with loads corresponding to 4% of the body weight, while the high group had 20-min sessions with loads corresponding to 6% of the body weight. The control group remained sedentary during the same period. During the experiment, three weight measurements were taken each week using a DG200 analytical scale (Digimed Analítica Ltd., São Paulo, Brazil), always at 9 AM, under the same evaluator. The average amount of food consumed was measured for each experimental group, also three times per week, during the training period. A 700 g quantity of feed was provided every two days in each cage, with the difference between the total amount inserted and what remained at the end of the period divided by the number of animals in the environment being used as the group’s average.

Before the start of the regimen and after the last day, prior to euthanasia, echocardiogram measurements were taken to estimate the cardiac function (FUJIFILM VisualSonics Inc., Toronto, ON, Canada), with the animals anesthetized using tribromoethanol to reduce motion artifacts. Then, after they resumed normality, 12 h of fasting was followed by the collection of 10 µL of blood through heparinized capillaries, deposited in 0.5 mL Eppendorf tubes with 20 µL of 1% sodium fluoride and frozen until the moment of glycemia analysis using the electroenzymatic method through the YSI 2700S biochemical analyzer (YSI Inc., Yellow Springs, OH, USA).

To avoid the acute effects of exercise, 72 h after the end of the training period, the animals were sacrificed via the administration of 10% ketamine (150 mg/kg) and 2% xylazine (15 mg/kg). Samples from the heart and gastrocnemius muscular tissues were collected, due to the heart’s heightened activity during any exercise and the gastrocnemius being activated during swimming [113], for the extraction of the necessary material and stored at −80 °C in RNALater (Invitrogen, Carlsbad, CA, USA) for use in the next stages.

**Molecular analysis:** For the analysis of the genes related to the circadian clock, total RNA was extracted using TRIzol (Invitrogen), according to the manufacturer’s recommendations. To cause cell lysis, all samples were homogenized in the Tissuelyser LT (Retsch GmbH, Haan, Germany; (QIAGEN GmbH, Hilden, Germany) device using two 5 mm, RNAse-free, steel spheres at 50 Hz, twice, for 5 min each. After extraction, DNAse I (Ambion^®^, Thermo Fisher Scientific, Austin, TX, USA) was used to eliminate the genomic DNA. The RNA samples were then quantified using the Qubit^®^ 1.0 fluorometer (Qiagen), with the RNA quality being evaluated by the Agilent 2100 Bioanalyser equipment with the Pico Labchip Kit (Agilent Technologies Inc., Santa Clara, CA, USA), generating the RNA Integrity Number (RIN). The samples were subsequently used for cDNA synthesis using the High-Capacity cDNA Reverse Transcription (Thermo Fisher Scientific) Kit, following the manufacturer’s instructions. Finally, qPCR reactions were performed using the SYBR (Thermo Fisher Scientific) method, with a standardized amount of 1 μg of cDNA for all samples, incubated in the QuantStudio^®^ 3 (ThermoFisher™) thermocycler. The genes of interest were *Clock*, *Arntl*, *Nr1d1*, *Per1*, *Per2*, *Cry1*, and *Cry2*, with 36B4 (*Rplp0*) as the endogenous control and the mice primers shown in Table 1 as guides.

The Ct (threshold cycle) value was used to estimate the specificity of the amplified products, with single visible peaks for each sample at the end of the run demonstrating no unspecific amplification. Triplicates were run on the same plate for each sample and, to avoid systematic differences within the PCR procedure, they were randomly distributed in the area. Negative controls were included, leading to the absence of magnification (Ct = 0) reliably being considered negative [114,115]. The 2^−ΔΔCt^ method was used to calculate the relative quantification of transcript levels, comparable by fold change, as described by Livak and Schmittgen [116].

**GLUT4 analysis:** As a complimentary analysis to accompany the glycemia measurements, the GLUT4 content in the plasmatic membrane of the gastrocnemius cells was also measured. For this purpose, first, the tissue was homogenized in the STE buffer with protease inhibitors (Sigma FAST, MilliporeSigma, Burlington, MA, USA) and then centrifuged serially, which was followed by a protein concentration measurement. Then, SDS-PAGE electrophoresis (Bio-America Inc., Toronto, ON, Canada) was performed, separating proteins by their molecular mass. The spectrum of ~54 kDa is expected for GLUT4, so that region is captured using the BIO-RAD system (Bio-Rad Laboratories Inc., Hercules, CA, USA) to transfer it to a nitrocellulose membrane (MilliporeSigma). The transference quality was evaluated by staining and a known molecular mass measurement. Finally, immunoblotting (MilliporeSigma) was performed with PBS-Tween as the blocker for non-specific binding, and the quantification of chemiluminescence emitted by the antigen–antibody reactions in the bands was performed by optical densitometry (ImageJ 1.53t).

**Statistical analysis:** Descriptive characteristics are presented as means and standard deviations unless otherwise noted. The Shapiro–Wilk test and Levene tests were used to verify data distribution nature and homogeneity, respectively. Gene expression associations were explored using principal component analysis and presented as a biplot according to the interventions (control, low, and high) and tissues analyzed (the gastrocnemius muscle and the heart). Furthermore, a heatmap was obtained for all genes according to each intervention group. A one-way ANOVA followed by Tukey’s post hoc test were performed to compare gene expression between groups. Body weight, the lactate threshold, and the maximum workload pre- and post-training were also analyzed by using a two-way ANOVA followed by Fisher’s post hoc test. *p* < 0.05 was considered significant. All analyses were performed using R Version 4.1.3 and RStudio 2022.02.0+443.

## 5. Conclusions

The investigation of the effects of different intensity regimens on the expression of clock genes evidenced that low-intensity exercise, performed at workloads below the anaerobic threshold, has more impact over the genomic regulation than higher-intensity exercise, especially in the gastrocnemius muscle. This is an interesting finding as high-intensity exercise is the usual choice to treat conditions involving defects in the circadian mechanisms and their secondary functions, such as obesity, diabetes, and hypertension. This contrasting finding may bring forth exciting novel approaches in circadian medicine, with diagnosis based on the identification of the anaerobic threshold and the prescription of lower-intensity exercise for improving physical capacity, heart condition, lipid metabolism, and other parameters.

## Figures and Tables

**Figure 1 ijms-26-08739-f001:**
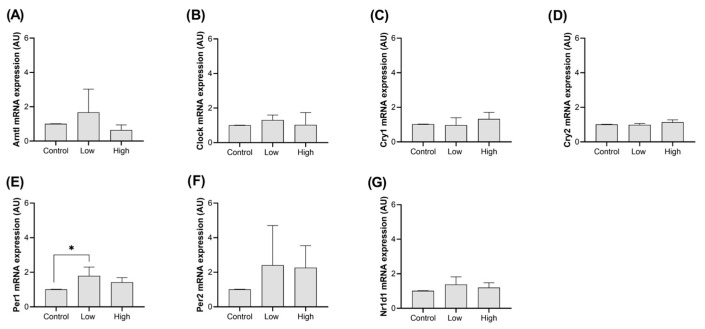
*Clock* gene expression in the heart. The Y axis represents the gene expression in Arbitrary Unit (AU), based on the average of the expression coefficients (e) for each group, with error bars indicating the standard deviation of each value. Each bar represents an experimental group: control, low-intensity exercise, and high-intensity exercise. Each plot represents a single clock gene, as follows: (**A**) *Arntl*, (**B**) *Clock*, (**C**) *Cry1*, (**D**) *Cry2*, (**E**) *Per1*, (**F**) *Per2*, and (**G**) *Nr1d1*. Statistical significance is indicated by asterisks (*), with their absence indicating no significant difference between groups.

**Figure 2 ijms-26-08739-f002:**
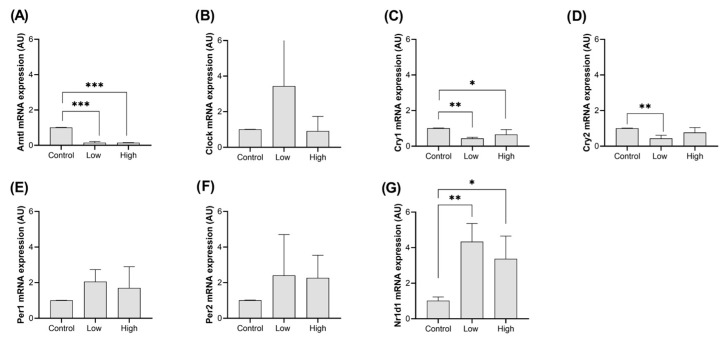
Clock gene expression in the gastrocnemius muscle. The Y axis represents the gene expression in Arbitrary Unit (AU), based on the average of the expression coefficients (e) for each group, with error bars indicating the standard deviation of each value. Each bar represents an experimental group: control, low-intensity exercise, and high-intensity exercise. Statistical significance is indicated by asterisks (*), with more asterisks denoting higher significance, and their absence indicating no significant difference between groups. Each plot represents a single clock gene, as follows: (**A**) *Arntl*, (**B**) *Clock*, (**C**) *Cry1*, (**D**) *Cry2*, (**E**) *Per1*, (**F**) *Per2*, and (**G**) *Nr1d1*.

**Figure 3 ijms-26-08739-f003:**
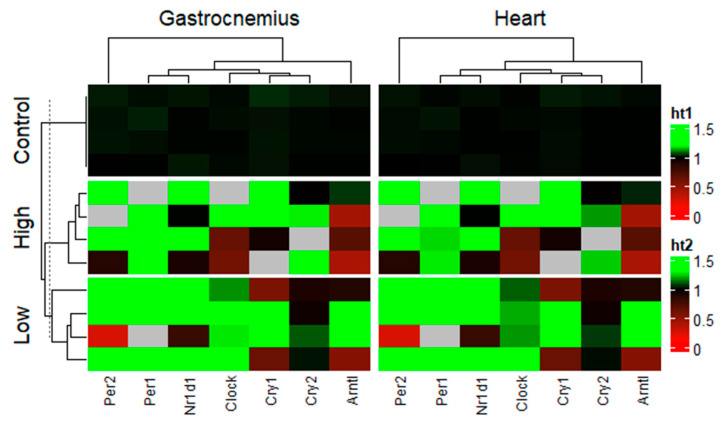
Heatmap and dendrogram clusters for gene expression according to tissues and interventions. Each square represents the individual expression of each clock gene for a single specimen in the group. Red heat (ht) indicates low expression, green heat (ht) indicates high expression, and black heat (ht) represents the control values.

**Figure 4 ijms-26-08739-f004:**
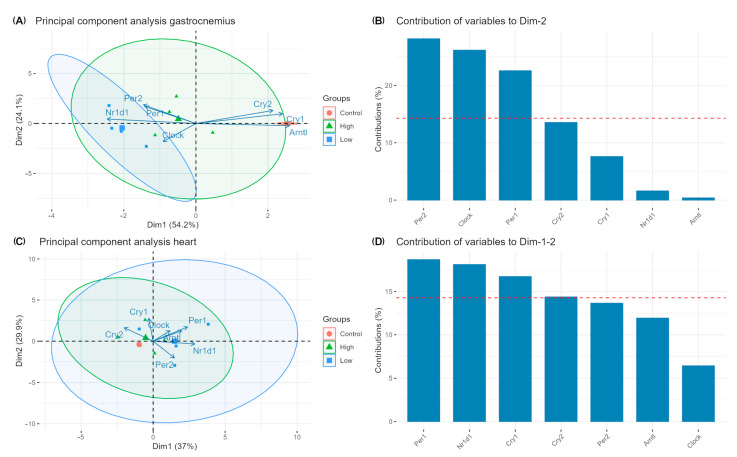
Principal components analysis (PCA) and contribution of variables for the observational variances. Ellipses are created by the K-means algorithm and represent clusters of data points. Arrows and columns indicate the contribution of each gene to the principal components. The dimensions (Dim) Dim1 and Dim2 represent the principal components that capture the most variance in the data, with their percentages showing the proportion of the total variance explained by each component, and the threshold for significant contribution being represented by the red dashed lines. Plots (**A**,**B**) refer to the gastrocnemius muscle, while plots (**C**,**D**) refer to the heart.

**Table 1 ijms-26-08739-t001:** Primers used in qPCR.

Gene	Accession No.	Sense Primer (5′-3′)	Antisense Primer (5′-3′)
36B4	NM_007475	GAGGAATCAGATGAGGATATGGGA	AAGCAGGCTGACTTGGTTGC
*Clock*	NM_007715	TTGCTCCACGGGAATCCTT	GGAGGGAAAGTGCTCTGTTGTAG
*Arntl*	NM_007489	GGACTTGCGCTCTACCTGTTCA	AACCATGTGCGAGTGCAGGCGC
*Nr1d1*	NM_145434	GGTGCGCTTTGCATCGTT	GGTTGTGCGGCTCAGGAA
*Per1*	NM_011065	GCGGGTCTTCGGTTAAGGTT	AGGCTCAGCTGGGATTTGG
*Per2*	NM_011066	CCAACACAGACGACAGCATCA	CTTCAACACCGCCTGGAGAT
*Cry1*	NM_007771	AGATCTTGGTAAGAGATTTGCTTAATGTAA	TGACTCTCAAAACTCTTGAGATTTATATCA
*Cry2*	NM_009963	AGCCCAGGCCAAGAGGAA	GTTTTTCAGGCCCACTCTACCTT

## Data Availability

The data presented in this study is available on request from the corresponding author.

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
