# Peer review of "Clock Gene Expression Modulation by Low- and High-Intensity Exercise Regimens in Aging Mice"

_ijms, 2025, doi:10.3390/ijms26178739_

Round 1

Reviewer 1 Report (New Reviewer)

Comments and Suggestions for Authors

This study aimed to investigate the differences in the expression of clock genes that regulate circadian rhythms in the gastrocnemius muscle and heart of aged mice after intervention with moderate-intensity and high-intensity aerobic swimming training. However, the current results are insufficient to support this intriguing research. My opinions are as follows:

  1. The presentation form of the results section is not conducive to readers' comprehension. It is suggested that the results be described point by point to highlight the logical hierarchy.
  2. In the experimental group design, each group contains only 4 aged mice. Given that significant individual differences may exist during the aging process, and the researchers failed to effectively evaluate the aging status of mice in each group, such a small sample size is not conducive to accurately presenting the experimental results.
  3. The results should present the differences in exercise capacity and cardiac function among mice in each group after aerobic swimming training of different intensities.

Author Response

Please see the attachment for our throughout response to your comments! Thank you very much for your insights, they were paramount to improving the article and we hope it is closer to your clearly high quality standards now!

Reviewer 2 Report (New Reviewer)

Comments and Suggestions for Authors

The peer-reviewed article is a structured and methodologically sound study devoted to the study of the effect of physical exercise of varying intensity on the expression of genes regulating circadian rhythms in aging mice.
The work is relevant in the context of studying circadian rhythm disorders, as well as searching for non-drug methods of correcting age-related disorders of daily rhythmicity.
The study addresses an important problem of age-related changes in circadian rhythms and their relationship with metabolic disorders. Physical exercise is considered as a potential tool for regulating the expression of clock genes, which is important for the prevention and therapy of age-associated desynchronosis and diseases.
The authors used standardized swimming protocols for mice with a clear division into low- and high-intensity groups. Then, a comprehensive analysis of the expression of key circadian cycle genes (*Clock, Arntl, Per1 / 2, Cry1 / 2, Nr1d1 *) was carried out in two types of tissues (heart and gastrocnemius muscle). The statistical analysis was performed correctly, using ANOVA and post-hoc tests.
The authors found that low-intensity exercise significantly affects gene expression in skeletal muscle (*Arntl, Cry1/2, Nr1d1*), while in the myocardium only Per1 expression changes. The hypothesis that exercise below the anaerobic threshold is more effective in modulating circadian genes than high-intensity exercise was confirmed.
The results may be useful for developing recommendations for physical activity in older adults with circadian rhythm disorders.
However, the article is not without its shortcomings. In particular, the study used only 12 mice (4 per group), which reduces statistical power, and all the mice were male. The most significant, but easily fixable, shortcoming is that it is not discussed how exactly changes in gene expression affect functional indicators (e.g., metabolism or cardiac function), and there are no data on behavioral or physiological markers of circadian rhythms (e.g., activity, body temperature). In addition, there is little discussion of the fact that swimming is a stressor for rodents, which could have influenced the results.
Thus, the article makes a significant contribution to the understanding of the effects of exercise on circadian rhythms in the context of aging. The data obtained indicate the advantage of low-intensity exercise for modulating the expression of clock genes, which opens up prospects for application in gerontology and sports medicine. After making the necessary edits, I recommend it for publication

Author Response

Please see the attachment for our through response to your comments! Thank you very much for the review, your insights were paramount for improving the article!

Round 2

Reviewer 1 Report (New Reviewer)

Comments and Suggestions for Authors

I have no comments at this time and agree to accept the manuscript.

This manuscript is a resubmission of an earlier submission. The following is a list of the peer review reports and author responses from that submission.

Round 1

Reviewer 1 Report

Comments and Suggestions for Authors

The reviewer suggests concentrating on these essential areas to enhance the manuscript.

1. The current abstract is misleading, as it is written in the style of a review article when in fact it describes a regular research article. The abstract requires significant revision to clearly communicate the study's novel aspects and underlying rationale, including whether the focus was on clock genes or exercise. The objective section is overly verbose, lacking a concise summary of the overall rationale. Additionally, the results section is incomplete, omitting crucial data and findings needed to adequately support the stated conclusions.

2. The current introduction contains broad, general information about clock genes that lacks direct relevance to the specific focus of this paper. The introduction should instead concentrate more narrowly on the molecular approach and other key topics examined in the context of exercise and aging.

3. The concluding introductory paragraph should clearly and concisely communicate the paper's importance, central aim, and the reader's interest in the study's approach.

4. The results section lacks a clear, logical structure, presenting the data in a disjointed and challenging manner. To enhance clarity, the authors should reorganize the information into a coherent, structured format.

5. The study found no significant relationship between clock gene expression and heart tissue, except for Per1 mRNA. The authors should evaluate their findings in Per1 knockout mice.

6. The study found that the gastrocnemius muscle exhibited a distinct mRNA expression pattern for clock genes compared to heart tissues. However, it remains unclear if this tissue-specific difference is related to exercise. The authors should provide further clarification on this finding. 

7. While the study raises intriguing questions, it fails to clearly elucidate the molecular underpinnings of the observed traits. Furthermore, the potential indirect effects of exercise on the heart and gastrocnemius muscle warrant a more comprehensive investigation into the molecular mechanisms driving these changes.

8. Experimental validation is crucial to confirm the observed gene expression changes. To ensure the reliability of the findings, the selected threshold should be validated using independent methods, such as western blot analysis and immunohistochemistry.

9. The authors should clearly identify the key strengths and limitations of their research, then propose compelling new avenues for future exploration in this field.

10. The figure legends are problematic, undermining their effectiveness. They overuse abbreviations that may confuse or alienate readers unfamiliar with the specialized terminology. Moreover, the legends do not provide the necessary details for readers to gain even a basic understanding of the experimental methods employed.

Comments on the Quality of English Language

Rephrasing sentences can enhance clarity and overall textual coherence. To achieve a polished, fluid narrative, it's highly advisable to collaborate with a native English speaker or professional language editing service.

Reviewer 2 Report

Comments and Suggestions for Authors

·         why methods after results and discussion?

·         did the mice exercise was conducted at a specific time of day. circadian rhythms are time-sensitive, the lack of time-of-day standardization could introduce variability in gene expression results.

·         Line 342: Why male only? sex differences in circadian rhyrm and exercise responses are well-documented.

·         Line 344: Sample size need to be clarified and also justified n =12? this should be also called feasibility study.

·         Line 377: heart and gastrocnemius muscles were collected justify liver or brain (e.g., suprachiasmatic nucleus), were not evaluated, potentially missing broader systemic effects.

·         mRNA expression was assessed, but the study does not quantify protein levels of clock genes or their functional outcomes this need to be explained. no behavioural parameters e.g. sleep/wake rhytm cycles, locomotor activity and no phsyciological e.g. temp and cardiac..

·         pls improve report of all statistical tests inclusing test values and p-values.

Reviewer 3 Report

Comments and Suggestions for Authors

This is an interesting research article with adequate novelty. Some points should be addressed.

- The 1st paragraph of the Introduction section should be enriched with more information on the topic.

- The 5th paragraph of the Introduction section is too long and not easily readable. This paragraph should be split into two smaller paragraphs.

- The resolution and the quality of Figures 1 and 2 should be improved.

- Again the resolution and the quality of Figure 4 should be improved.

- At the end of the Discussion section, the authors should add a paragraph with the strengths and the limitations of their study.

Round 2

Reviewer 1 Report

Comments and Suggestions for Authors

While the central idea is novel, this reviewer has significant concerns regarding the study design and data interpretation. The research findings presented are preliminary and require further validation before publication can be recommended. Additional work is needed to robustly support the conclusions outlined in the manuscript.

The revised paper addresses most of my comments, but lacks molecular verification or cellular or molecular-level insights (Western blotting or Immunohistochemistry).

Reviewer 2 Report

Comments and Suggestions for Authors

i am satisfied with all changes 

however i want the changes to be refelected in the paper readers will have same quetions as i did 

e.g. 

Line 377: heart and gastrocnemius muscles were collected justify liver or brain (e.g., suprachiasmatic nucleus), were not evaluated, potentially missing broader systemic effects. Since this was a follow-up experiment, we had limited access to samples, only having heart and gastrocnemius tissue samples to work with. Indeed, however, liver, brain and even blood samples would be highly enriching for the project.

so please add these to dicussion

Reviewer 3 Report

Comments and Suggestions for Authors

The authors have significantly improved the manuscript.
